# Effect of Feeding Wet Feed or Wet Feed Fermented by *Bacillus licheniformis* on Growth Performance, Histopathology and Growth and Lipid Metabolism Marker Genes in Broiler Chickens

**DOI:** 10.3390/ani11010083

**Published:** 2021-01-05

**Authors:** Ahmed A. Saleh, Mustafa Shukry, Foad Farrag, Mohamed M. Soliman, Abdel-Moneim Eid Abdel-Moneim

**Affiliations:** 1Department of Poultry Production, Faculty of Agriculture, Kafrelsheikh University, Kafrelsheikh 33516, Egypt; 2Department of Physiology, Faculty of Veterinary Medicine, Kafrelsheikh University, Kafrelsheikh 33516, Egypt; mostafa.ataa@vet.kfs.edu.eg; 3Department of Anatomy and Embryology, Faculty of Veterinary Medicine, Kafrelsheikh University, Kafrelsheikh 33516, Egypt; foad.farrag@yahoo.com; 4Clinical Laboratory Sciences Department, Turabah University College, Taif University, P.O. Box 11099, Taif 21944, Saudi Arabia; mmsoliman@tu.edu.sa; 5Biochemistry Department, Faculty of Veterinary Medicine, Benha University, Benha 13737, Egypt; 6Biological Application Department, Nuclear Research Center, Atomic Energy Authority, Abu-Zaabal 13759, Egypt; aeabdelmoneim@gmail.com

**Keywords:** performance, dry feed, fermented wet feed, intestinal morphology, gene expression, broilers

## Abstract

**Simple Summary:**

Heat stress is an abiotic environmental factor that affects poultry performance and results in several physiological, immunological and behavioral changes in birds. Previous works have illustrated that wet feeding has been considered as a useful agent to improve birds’ ability to resist high ambient temperatures and to enhance our understanding of feed consumption limitations when broilers are fed dry diets. In broilers, wet feeding enhances nutrient use. This research reveals the impacts of three feeding methods (dry feed, wet feed or wet feed fermented with *Bacillus licheniformis*) on the growth efficiency, intestinal histomorphometry and gene expression of the lipid metabolism- and growth-related genes of broiler chickens. Our findings confirm improved growth and digestibility for fermented wet feed compared with control and wet feed diets; however, wet feed has a negative effect on performance compared with the control group (dry feed). Additionally, fermented wet feed caused a reduced hepatic gene expression of peroxisome proliferator-activated receptor γ (*PPARγ*) and increased that of fatty acid synthase (*FAS*).

**Abstract:**

The present study evaluated the effect of three feeding methods (dry feed, wet feed or wet feed fermented with *Bacillus licheniformis*) on the growth performance, intestinal histomorphometry and gene expression of the lipid metabolism- and growth-related genes of broiler chickens. A total of 360 one-day-old Cobb-500 broiler chicks were randomly allotted into three groups containing four replicates with 30 birds each. The first group (control) was fed a dry mash basal diet. The second and third groups were fed wet feed and fermented wet feed. The final body weight and weight gain were reduced (*p* < 0.01) in the wet feed group, while they did not differ between the fermented wet feed and dry feed groups. Feed intake was not altered, and feeding on wet feed significantly (*p* < 0.01) increased the feed-to-gain ratio compared to the remaining groups. No differences between the three feeding methods in carcass characteristics, blood biochemistry and nutrient digestibility were observed except for crude protein digestibility, which was increased (*p* < 0.01) in the fermented wet feed group. Duodenal and ileal villi heights were elevated in birds fed fermented wet feeds, while crypt depth was not altered. The expression fold of *IGF-1*, *GH* and m-*TOR* genes in the pectoral muscle of birds fed wet feed was decreased (*p* < 0.05), while myostatin gene expression was elevated. Feeding on wet feed reduced the hepatic gene expression of *PPARγ* and increased that of *FAS*. In conclusion, wet feed negatively affected the broiler chickens’ efficiency under heat stress; however, fermenting the wet feed with *Bacillus licheniformis* improved feed utilization and birds’ performance compared to the dry feed group.

## 1. Introduction

Under summer conditions, elevated ambient temperature, as an abiotic environmental factor, impairs poultry performance and evokes several physiological, immunological and behavioral changes in birds [1,2]. Biotic factors such as nutrition and infectious diseases can act as environmental stressors independently and often elevate the risks of abiotic factors synergistically [3,4]. Feeding practices influence heat production by birds and their feed consumption, which is also depressed in a heat stress environment.

Wet feeding has been considered as a valuable tool to enhance birds’ capability to endure high ambient temperatures and to increase our understanding of feed consumption limitations when broilers are fed dry diets [5,6,7,8]. Dei and Bumbie (2011)reported that wet feeding improved feed intake (FI) and reduced the impact of heat stress on broilers reared in hot tropical conditions. Nevertheless, there were contradictions in the investigation’s results concerning the effect of wet feeding on poultry efficiency [9]. Afsharmanesh et al. (2016), Tabeidian et al. (2015) and Atapattu and Sudusinghe (2013) showed significant improvements in the FI, feed conversion ratio (FCR) and daily weight gain (WG) of broilers when fed on wet feed [10,11,12]. However, others did not report noticeable changes in the WG and FCR of broilers, Muscovy ducks and geese as a result of wet feeding compared to dry feeding [6,7,13,14,15]. Thus, the further processing of wet feeds or supplement inclusion is needed to ensure that beneficial effects on growth performance are attained.

*Bacillus* spp. seem to be an alternative to antimicrobial growth promoters for improving animals’ health and performance. Within this regard, Bacillus licheniformis have the ability to sporulate, thereby making them stable during thermal treatment of feed and resistant to enzymatic digestion along the gastrointestinal tract (GIT) [16]. Previous studies reported that Bacillus licheniformis supplementation in broilers’ drinking water or diets promoted growth performance by enhancing the body weight and feed conversion ratio of broilers [6,7,8,9,10,11,12].

Feed fermentation before feeding refers to the conversion of carbohydrates to carbon dioxide, alcohol and organic acids with or without specific inocula [6,17]. The fermentation of wet feed in vitro can improve crude protein and organic acid digestibility [18]. Scholten et al. (1999) found that the enumeration of gut coliform bacteria and gastric pH in pigs was reduced when fed on fermented wet feed [19]. Feeding broilers fermented wet feed could be a superior means to enhance the use of feed nutrients and improve efficiency compared to wet feeding [6]. However, despite the advantages of wet feeding and because fermentation commences quickly when water is mixed with feed, the potential for microbial proliferation in wet feed is a primary concern. This may result in birds being affected by direct microbial infection or harmful toxins from contaminated feed [20]. Consequently, inoculating the wet feed before fermentation with probiotic bacteria could maximize the beneficial effect of fermented wet feed along with gaining the benefits of the probiotics themselves on both the feed and the host.

Earlier studies have investigated the favorable effect of fermented wet feed in tests conducted on pigs; however, the information available on poultry, particularly on broilers, is restricted. Therefore, the current study aimed to elucidate the impact of dry, wet and fermented wet feeds on the growth efficiency, intestinal histomorphometry and gene expression of lipid metabolism- and growth-related genes in heat-stressed broiler chickens.

## 2. Materials and Methods

### 2.1. Birds and Experimental Diets

The study was approved by the ethics committee of the local experimental animals care committee and conducted in accordance with the guidelines of Kaferelsheikh University, Egypt (Number 4/2016 EC). In total, 360 one-day-old Cobb-500 male broiler chicks were purchased from a local hatchery, weighed upon arrival and randomly allocated into three experimental groups containing four replicates with 30 birds each. The experiment lasted until the chicks reached 35 days of age. Each replicate pen was littered with wood shavings of 7 cm in thickness with a surface area of 2 m^2^. Birds were kept at 33–30 °C during the first 14 days, which was progressively reduced by 3 °C until the natural ambient temperature of summer conditions was reached. The room temperature and relative humidity were recorded from 10 a.m. to 10 p.m. and ranged from 32 to 28 °C and 46.5% to 63.0%, respectively. Birds were offered water and corn–soybean meal-based starter (1 to 21 D) and grower (22 to 35 D) diets ad libitum (Table 1). Experimental diets were formulated according to the recommendation of NRC (1994) [21]. The dietary handlings were as follows: (1) dry mash basal diet, considered as the control (dry feed group), (2) a wet feed group, which received a fresh wet mash diet prepared by mixing the diet with tap water at a ratio of 1:1.4 (wt/wt) and (3) a fermented wet feed group, which was fed wet feed fermented by *Bacillus licheniformis*.

The fermentation of wet feed was prepared by modifying the method reported by Heres et al. (2003) [22]. In brief, one kilogram of wet feed with the same feed-to-water ratio mentioned above was inoculated with 5 g B-Act^®^ (*Bacillus licheniformis* DSM 28710 at 1.6 × 10^9^ CFU/kg of feed). The mixture was incubated for three days at 30 °C in sterile glass jars. The pH was monitored after the fermentation process as a quality control procedure until it reached 4. The prepared fermented wet feed was used within seven days and stored at 4 °C until that time.

### 2.2. Growth Performance

Body weight (BW) was documented at the beginning and the end (35 days of age) of the experimental period. Feed intake was estimated by measuring the amount of feed offered and residue left, and the feed conversion ratio (FCR) was calculated as g feed/g gain. At the end of the experiment, two birds per replicate (eight birds for each group) in the same body weight were slaughtered. The hot carcass, liver, gizzard, heart, spleen and abdominal fats were collected, weighed and evaluated in proportion to the live BW. Carcass yield and dressing (%) were calculated as described by Abd El-Moneim et al. (2020) and Abdel-Moneim et al. (2020) [23,24]. The average weight of two thighs and breast weight (breast muscles with the sternum) were recorded and evaluated in proportion to the live BW. Tissue samples were collected from the liver and breast muscle, dipped immediately in liquid nitrogen and kept at −80 °C until real-time PCR analysis.

### 2.3. Nutrients Digestibility

At 32 days of age, eight male broiler chicks from each group were weighed and housed in metabolic cages individually. After a 24-h adaptation period, feces samples were collected for three consecutive days. Feed and water were offered ad libitum by using fixed containers. Birds were also weighed after the collection period to ensure that they were maintaining their weight. The proximate analysis of crude protein (#954.01), ether extract (#920.29) and crude fiber (#978.10) in diets and dried excreta was conducted following the (AOAC 2003) [25]. Trichloroacetic acid was used to estimate the fecal nitrogen following the procedure described by Jacobsen et al. (1960) [26].

### 2.4. Blood Biochemistry

At the end of the experiment, blood samples were obtained in labeled tubes from eight birds per group, left to clot and centrifuged at 3500× *g* for 15 min. Sera samples were separated and stored at −20 °C until biochemical analysis. Total protein, albumin, globulin, aspartate aminotransferase (AST), alanine aminotransferase (ALT), glucose, total cholesterol (TC), triglycerides (TG) and very low density lipoprotein (VLDL) were spectrophotometrically assessed (Spectronic 1201; Milton Roy, Ivyland, PA, USA) using commercial kits (Cell Biolabs Inc., San Diego, CA, USA) according to the manufacturer’s instructions.

### 2.5. Morphometry of Intestine

At the end of the feeding experiment, eight birds from each group were randomly selected and slaughtered for sampling. The tissue samples from the duodenum, ileum and cecum were fixed in Bouin’s solution for 18–24 h. After fixation, the samples were dehydrated by using ascending concentrations of ethyl alcohol (70% to absolute alcohol), then cleared in xylene and prepared for histological investigations. Sections of 4–5 µm thickness were stained with hematoxylin and eosin for general morphometry, according to (Bancroft and Layton 2019) [27]. The length of intestinal villi in addition to crypt depth was measured by using image analysis software (NIH, Bethesda, MD, USA.). A total of eight random villi and villus-associated crypts from eight intestinal cross-sections were selected, and the average was calculated.

### 2.6. Real-Time Polymerase Chain Reaction (RT-PCR)

The gene expression of lipoprotein lipase (*LPL*), peroxisome proliferator-activated receptor γ (*PPARγ*) and fatty acid synthase (*FAS*) in hepatic tissues and myostatin, myogenin, mammalian target of rapamycin (*mTOR*), insulin-like growth factor 1 (*IGF-1*) and growth hormone (*GH*) in breast muscle were determined with RT-PCR. In short, the total RNA of approximately 100 mg of the tissue was extracted using TRIzol reagent (Invitrogen, Life Technologies, Carlsbad, CA, USA) and Nanodrop for quantification. RNA samples of 1.8 or more A260/A280 were used for the synthesis of DNA using a cDNA (Fermentas, Waltham, MA, USA) synthesis kit. The SYBR green master mix and the primers in Table 2 with the household gene (*GAPDH*) were added to amplify cDNA. Amplification data were analyzed using 2^−ΔΔT^ methods [28].

### 2.7. Statistical Analysis

Collected data were analyzed using a one-way analysis of variance, with the general linear model procedure, by the statistical analysis system SPSS 19 (2018). One-way ANOVA was applied to determine the effects of wet feed, fermented wet feed, when birds were the statistical units for body weight and pens used for feed intake. Tukey’s multiple comparison test was used. (≠) indicates no significant changes, (*) indicates (*p* < 0.05), (**) indicates (*p* < 0.01). Asterisk located above the columns indicates significance between different treated groups.

## 3. Results

### 3.1. Growth Performance

The results of the growth performance of broiler chickens fed dry, wet or fermented wet feeds are presented in Table 3. The final BW and WG were decreased (*p* < 0.01) in the wet feed group, while no differences in their values were noticed between fermented wet feed and dry feed groups. Birds fed wet feed recorded a numerical increase (*p* = 0.055) in daily feed intake compared to those fed dry or fermented wet feeds. Feeding on wet feed significantly (*p* < 0.01) increased the FCR compared to the remaining groups. The apparent digestibility coefficients of the ether extract and crude fiber of broilers were not significantly different for the three forms of feed. Crude protein digestibility was not altered in broilers fed dry or wet feeds, while it was increased (*p* < 0.01) in those fed fermented wet feed.

The impact of dry, wet and fermented wet feeds on the carcass characteristics of 35-day-old broilers is shown in Table 4. Percentages of dressing, carcass yield, liver, gizzard, heart, spleen, abdominal fat, breast and thigh were not significantly altered between the experimental groups.

### 3.2. Blood Biochemistry

Feeding on dry, wet or fermented wet feeds did not influence the values of serum proteins, hepatic enzymes, glucose and lipid profiles of broiler chicks at 35 days of age (Table 5).

### 3.3. Morphometry of Intestine

As depicted in Figure 1, Figure 2 and Figure 3, a normal intestinal structure was observed in the duodenum, ileum and cecum of broilers fed dry, wet and fermented wet feeds. The villus height (VH) was elevated in the duodenum and ileum of birds fed fermented wet feeds in relation to those fed dry or wet feeds, while crypt depth (CD) was not altered (Figure 4). The VH/CD ratio was also increased in the fermented wet feed group compared to the remaining groups.

### 3.4. Expression of Growth- and Lipid Metabolism-Related Genes

In the pectoral muscle samples, the mRNA expressions of *IGF-1*, *GH* and *mTOR* were decreased (*p* < 0.05) in the wet feed group and numerically increased in the fermented wet feed group compared to the control (Figure 5). The expression fold of myogenin was not significantly different between the experimental groups, while myostatin gene expression was higher (*p* < 0.05) in the wet feed group. Feeding on wet feed reduced the hepatic gene expression of *PPARγ* and increased that of *FAS*, while *LPL* gene expression was not altered compared to the remaining groups.

## 4. Discussion

One of the major objectives of the current study was to evaluate the impacts of feeding fermented feed on performance. The results of the present study reveal that wet feed reduced BW and WG and increased FI and FCR values. These findings agree with those of Liu et al. (2019), who noticed insignificant decreases in BW and WG and substantial rises in FI and FCR in wet-fed geese [14]. Furthermore, Akinola et al. (2015), Emadinia et al. (2014) and Farghly et al. (2018) described insignificant changes in the WG and FCR of broilers and Muscovy ducks as a result of wet feeding compared to dry feeding [6,7,13]. Scott and Silversides (2003) also reported an increase in the daily FI of broilers that received wet feeds [29].

On the other hand, others have reported significant improvements in the BW, FI and FCR of broilers when fed on wet feed [10,11,13,30]. In this study, the marked differences in growth performance between wet feed and dry feed can be ascribed to the variations in the digesta passage rate through the gastrointestinal tract, which increased in the case of wet feeding [31,32]. This increase in the passage rate of digesta reduces the contact time between nutrients and digestive enzymes, microbial populations and absorptive surfaces, which are almost always accompanied by increased FI and decreased nutrient utilization and consequently lower weight gain. Our explanation agreed with Scott (2002) and Silversides, as the authors stated that the ability of broilers fed wet feed to utilize feed was decreased compared to those fed on dry feed [29]. From another perspective, the reduction in the performance of wet-fed broilers in this study might also be attributed to the fermentation of wet feed with unwanted microbes, which commences rapidly when water is mixed with feed, particularly under high ambient temperatures. The potential for undesired microbial proliferation in wet feed may lead to the direct microbial infection of birds or the risk of harmful toxins from contaminated feed, which negatively affect weight gain [19].

The fermentation of wet feed by *Bacillus licheniformis* addressed the previous negative impact of wet feed since the BW and WG of fermented wet-fed broilers were higher than those of wet-fed birds and similar to those of dry-fed birds. These findings agree with previous results in the literature [6,33,34]. It has been described that the administration of *B. Licheniformis* in broilers’ drinking water or diets, as probiotics, or its use in the fermentation of dietary materials, promoted the growth performance of these birds. Yasar, S.; Forbes and Liu et al. (2012) Gong et al. (2018) demonstrated that *B. Licheniformis* supplementation in drinking water or in diets enhanced the BW and WG of broilers [35,36]. *B. Licheniformis*-fermented products improved the growth efficiency parameters of broiler chickens [37]. Dietary fermentation with *B. Licheniformis* can increase organic acid concentrations, which reduces pH to almost 4, as shown in the present study and following Akinola et al. (2015) and Canibe et al. (2007) [6,38]. Due to the reduction in pH and the competitive exclusion mechanism, the growth of pathogens in the diet and in the gut is inhibited [1,39,40]. Gong et al. (2018) reported that lipase, amylase, total protease and trypsin activities in the duodenal contents were enhanced when broilers were treated with *B. Licheniformis* [36]. Additionally, the diets fermented with *B. Licheniformis* not only contained probiotic bacteria but also showed antibacterial cyclic lipopeptide derived from *B. Licheniformis*. Taken together, the previous findings indicate the positive effects of *B. Licheniformis* and *B. Licheniformis*-fermented diets on the utilization of nutrients and growth performance of broiler chickens.

In this study, neither dry, wet nor fermented wet feeds affected the carcass traits of broiler chickens at 35 days of age. These findings are in line with those presented in [6,7,12,41]. Liu et al. (2019) demonstrated that wet feeding does not significantly affect dressed, breast meat, leg meat, liver, heart and abdominal fat percentages in geese [14]. The carcass characteristics of Muscovy ducks fed wet feed were also not influenced [7]. Akinola et al. (2015) reported that the percentages of the cut-up parts and relative organ weight of broilers did not differ when fed on dry, wet or fermented wet feeds [6].

Conversely, Tabeidian et al. (2015) observed a significant decrease in abdominal fat percentage in wet-fed broilers [11]. Emadinia et al. (2014) and Afsharmanesh et al. (2006) observed that heart and liver relative weights were increased in wet-fed broilers compared with dry-fed ones [5,13]. The authors attributed the increase in liver and heart weights to the rise in the activity associated with the increase in BW resulting from the elevation in the feed intake of broilers that received a wet feed. In the present study, wet feeding or fermented wet feeding did not promote BW and thus had no impact on liver and heart growth.

The apparent digestibility of ether extract and crude fiber was not significantly different between the feeding methods, while crude protein digestibility was improved in broilers fed fermented wet feed. These results concur with the findings of Liu et al. (2019) who reported insignificant differences in the dry matter and ether extract digestibility of geese fed dry or wet feed [14]. The same results were reported in broilers [6]. The lack of significance for nutrient digestibility between wet feed and dry feed groups despite the increase in FI in the wet feed group may be due to the enhanced passage rate of digesta, which reduces digestion and absorption time, which may explain the lack of difference for nutrient retention [5]. The improvement in protein digestibility in fermented wet feed has been previously reported [17]. It could be ascribed to the ability of *B. Licheniformis* to improve total protease and trypsin activities in the duodenal contents of broilers [35].

Feeding on dry, wet or fermented wet feeds did not substantially alter any of the measured blood biochemical parameters. Consistent with our results, Afsharmanesh et al. (2010) stated that feed form (wet and dry) did not affect the concentrations of blood glucose, HDL or total cholesterol [20]. Farghly et al. (2018) also revealed that total serum protein, albumin, globulin, AST, ALT, glucose and total cholesterol were not affected in growing Muscovy ducklings subjected to dry or wet feeding [7]. Furthermore, feeding broilers on *B. licheniformis*-fermented products did not affect their blood concentrations of AST, ALT, HDL, LDL, glucose, triglycerides and total cholesterol.

The results of the present study show a significant elevation in duodenal and ileal VH and VH/CD ratios in birds fed fermented wet feed. In contrast, no significant differences between dry and wet feeding were observed. These findings are consistent with earlier studies conducted in pigs, which indicated improvements in gut morphology as a result of fermented wet feeding [4,42,43]. Yasar and Forbes (2000) found a substantial rise in the VH and CD of broilers fed wet feed compared to those fed dry feed [31]. In contrast, Akinola et al. (2015) did not notice any significant changes in the histomorphometry of broilers fed on dry, wet or fermented wet feeds [6].

Protein metabolism plays an important role in the regulation of the mass of skeletal muscles, and it has been reported that *mTOR* signaling plays a fundamental role in the regulation of protein synthesis and degradation [44,45]. Moreover, muscle growth is regulated by *IGF-1* and *GH* via the *GH*–*IGF-1* axis, while the primary function of myostatin is believed to be the negative regulation of muscles [46]. To the best of our knowledge, the present study demonstrated, for the first time, the effect of dry, wet or fermented wet feeds on the gene expression of growth-related genes. In this study, muscle *IGF-1*, *GH* and *mTOR* expressions were boosted in the fermented wet feed group and decreased in the wet feed group. The expression fold of myostatin was upregulated in the wet feed group. Variations in the expression of growth-related genes between wet feed and fermented wet feed groups might be attributed to the availability of nutrients required to trigger the genetic potential of broiler chickens for growth. The decreased passage rate of digesta in the fermented wet feed group increases the digestion and absorption time, which in turn increases nutrient retention [5]. The availability of adequate quantities of nutrients elevated metabolic rates, thereby stimulating the production of growth hormone-releasing hormone from the hypothalamus to promote *GH* amplification [2,16,47]. In addition, the release of *IGF-1* is induced by the enhanced expression of *GH* [48]. The combination of the increased expressions of *IGF-I* and *GH* could result in the increased growth performance of broilers.

Gene expression in the liver plays an essential role in altering the digestive capability due to its ability to change the enzyme capacity of relevant metabolic pathways [49]. *FAS* is a rate-limiting lipogenesis enzyme that determines chickens’ ability to synthesize the fatty acids deposited in their body [50]. *PPARγ* is strongly correlated with abdominal fat deposition by regulating adipocyte differentiation and lipid metabolism [51,52,53,54]. *LPL* is necessary for fatty acid uptake by adipose tissue due to its ability to hydrolyze the triglycerides of LDL and chylomicrons to produce glycerols and free fatty acids [55,56,57]. In the present study, feeding on wet feed reduced the hepatic gene expression of *PPARγ* and increased that of *FAS*, while *LPL* gene expression was not altered when compared to the remaining groups. The phenomenon can be related to the change to the diet fermented by *B. licheniformis,* which can improve the digestion and nutrient utilization in wet feed [58,59,60]. In line with our findings, Zhou et al. (2016) found that the dietary administration of *B. licheniformis* upregulated the expression of *FAS* and *PPARγ* in hepatic tissues [61].

## 5. Conclusions

Our results show that wet feed has a negative impact on performance compared with dry feed; however, the fermentation of wet feed by *B. Licheniformis* addressed the disadvantages of wet feeding and improved protein digestibility. The use of fermented wet feed is more appropriate than the use of dry or wet feed as a potential approach to provide adequate nutrients to benefit from the genetic potential of broiler chickens’ growth.

## Figures and Tables

**Figure 1 animals-11-00083-f001:**
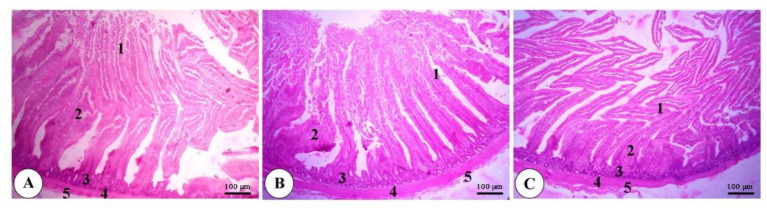
Representative photomicrograph of the duodenum of a broiler of the dry feed (**A**), wet feed (**B**) and fermented wet feed (**C**) groups showing the simple columnar epithelium of the tunica mucosa (1), connective tissue core of intestinal villi (2), intestinal glands (3), tunica muscularis (4) and tunica serosa (5). H&E, bar = 100 µm. (*n* = 8).

**Figure 2 animals-11-00083-f002:**
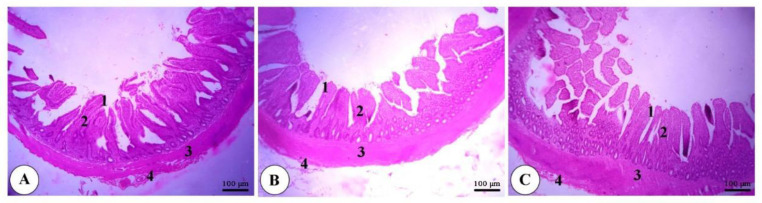
Representative photomicrograph of the ileum of a broiler of the dry feed (**A**), wet feed (**B**) and fermented wet feed (**C**) groups showing the simple columnar epithelium of the tunica mucosa (1), connective tissue core of intestinal villi (2), tunica muscularis (3) and tunica serosa (4). H&E, bar = 100 µm. (*n* = 8).

**Figure 3 animals-11-00083-f003:**
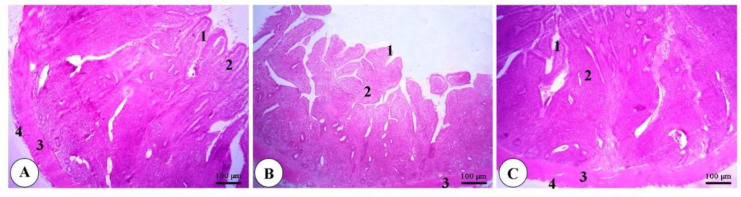
Representative photomicrograph of the cecum of a broiler of the dry feed (**A**), wet feed (**B**) and fermented wet feed (**C**) groups showing the simple columnar epithelium of the tunica mucosa (1), lamina propria with extensive deposits of lymphatic cells (2), tunica muscularis (3) and tunica serosa (4). H&E, bar = 100 µm. (*n* = 8).

**Figure 4 animals-11-00083-f004:**
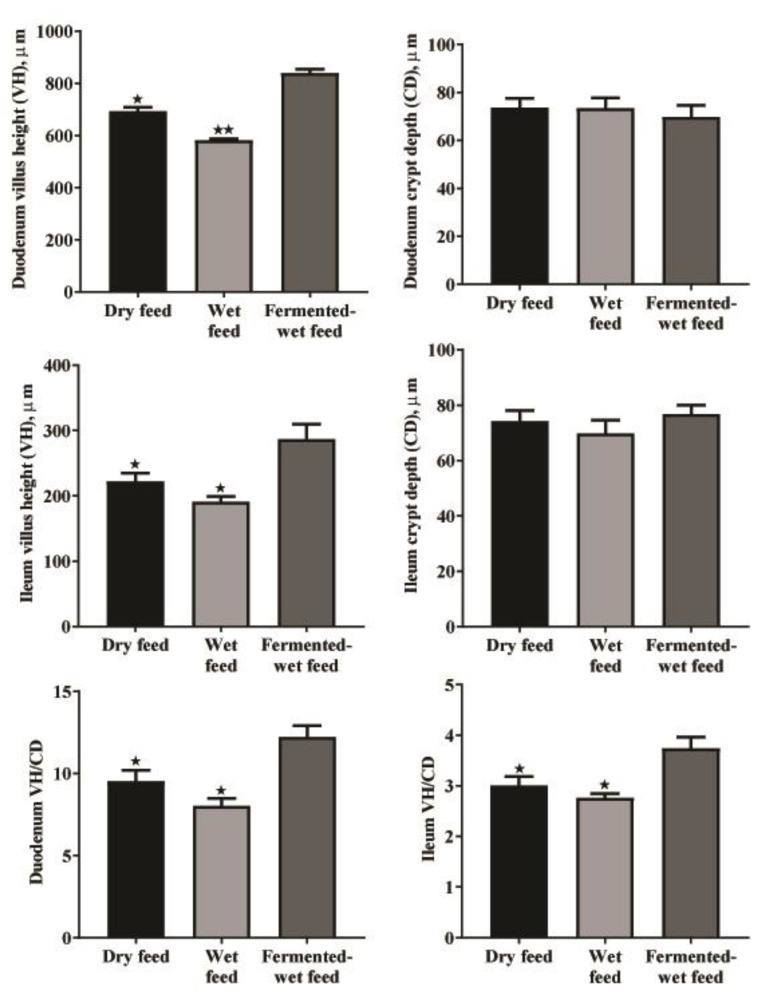
Duodenal and ileal histomorphometry of 35-day-old broiler chickens as affected by dry, wet and fermented wet feeds. Data are presented as the mean values with their standard errors. (*) indicates (*p* < 0.05), (**) indicates (*p* < 0.01). Asterisk located above the columns indicates significance between different treated groups and the fermented wet feed group. (*n* = 8).

**Figure 5 animals-11-00083-f005:**
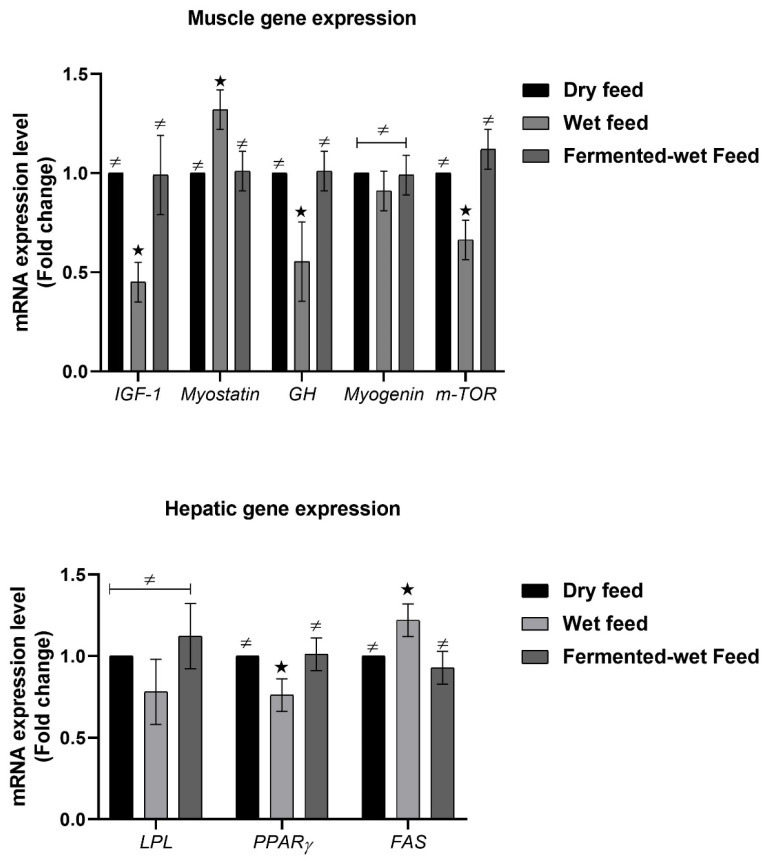
Reverse transcription-polymerase chain reaction (RT-PCR) validation of hepatic lipoprotein lipase (*LPL*), peroxisome proliferator-activated receptor γ (*PPARγ*) and fatty acid synthase (*FAS*), and muscular myostatin, myogenin, mammalian target of rapamycin (*mTOR*), insulin growth factor 1 (IGF1) and growth hormone (*GH*) of 35-day-old broiler chickens as affected by dry, wet and fermented wet feeds. Data are presented as the mean values with their standard errors. (≠) indicates no significant changes, (*) indicates (*p* < 0.05). Asterisk located above the columns indicates significance between different treated groups and the wet feed group. (*n* = 8).

**Table 1 animals-11-00083-t001:** Ingredients and calculated chemical composition of the basal diet.

Ingredients	Starter (1–21 Day)	Grower (22–35 Day)
Yellow corn, %	54.03	58.98
Soybean meal (44%), %	34.50	29.50
Corn germ (62%), %	5.50	5.50
Soya oil, %	1.80	2.30
Limestone, %	1.08	0.95
Di-Calcium Phosphate, %	2.00	1.75
Premix ^1^, %	0.30	0.30
NaCl, %	0.30	0.30
L-lysine, %	0.29	0.24
DL-Methionine, %	0.20	0.18
Calculated composition ^2^ (%)		
Metabolizable energy (ME, kcal kg^−1^)	3001	3180
Crude protein	23.12	20.99
Calcium	0.99	0.89
Potassium	0.54	0.52
Available Phosphorus	0.51	0.46
Digestible methio + Cys	0.93	0.89
Digestible methionine	0.59	0.52
Digestible lysine	1.43	1.24
Digestible arginine	1.25	1.07
Digestible tryptophan	0.19	0.17

^1^ Provides each kg of diet: Vit. A: 12,000 IU, Vit. D_3_: 5000 IU, Vit. E: 130.0 mg, Vit. K_3_: 3.61 mg, Vit. B_1_: 3.0 mg, Vit. B_2_: 8.0 mg, Vit. B_6_: 4.95 mg, Vit. B_12_: 0.17 mg, Niacin: 60.0 mg, Folic acid: 2.08 mg, D-Biotin: 200.0 mg, calcium D-Pantothenate: 18.33 mg, Copper: 80.0 mg, Iodine: 2.0 mg, Selenium: 150.0 mg, Iron: 80.0 mg, Manganese: 100.0 mg, Zinc: 80.0 mg, Cobalt: 500.0 mg. ^2^ Calculated according to NRC (1994).

**Table 2 animals-11-00083-t002:** Primers sequences and target genes for SYBR green RT-PCR.

Gene ^1^	Forward	Reverse	Accession Number
*IGF-1*	CATTTCTTCTACCTTGGC	TCATCCACTATTCCCTTG	M32791
*mTOR*	CCAGGATTCTTCGGACTA	CCATCACAAACCCTTATT	XM_417614
*Myostatin*	GGGACGTTATTAAGCAGC	ACTCCGTAGGCATTGTGA	NM 001001461
*GH*	CACCACAGCTAGAGACCCACATC	CCCACCGGCTCAAACTGC	HE608816
*Myogenin*	GCGGAGGCTGAAGAAGGT	AGGCGCTCGATGTACTGG	NM_204184.1
*LPL*	TTGGTGACCTGCTTATGCTA	TGCTGCCTCTTCTCCTTTAC	NM_205282
*PPARγ*	TCGCATCCATAAGAAAAGCA	CTTCTCCTTCTCCGCTTCGT	NM_001001460.1
*FAS*	CCAACGATTACCCGTCTCAA	CAGGCTCTGTATGCTGTCCAA	J03860
*GAPDH*	GGTGAAAGTCGGAGTCAACGG	CGATGAAGGGATCATTGATGGC	NM_204305

^1^*IGF-1*, insulin-like growth factor 1; *mTOR*, mammalian target of rapamycin; *GH*, growth hormone; *LPL*, lipoprotein lipase; *PPARγ*, peroxisome proliferator-activated receptor γ; *FAS*, fatty acid synthase; *GAPDH*, glyceraldehyde 3-phosphate dehydrogenase.

**Table 3 animals-11-00083-t003:** Effect of dry, wet and fermented wet feeds on growth performance and digestibility of broiler chickens at 35 days of age.

Item	Dry Feed	Wet Feed	Fermented Wet Feed	SEM ^1^	*p*-Value
Initial body weight ‘g’	49.32	49.35	49.31	0.128	0.991
Final body weight ‘g’	1842.6 ^a^	1765.0 ^b^	1814.5 ^a^	9.715	<0.001
Weight gain ‘g·bird·day^−1^’	51.23 ^a^	49.03 ^b^	50.43 ^a^	0.276	<0.001
Feed intake ‘g·bird·day^−1^’	77.93	81.60	79.38	0.646	0.055
Feed conversion ratio ‘g feed·g gain^−1^’	1.52 ^b^	1.66 ^a^	1.57 ^b^	0.075	<0.001
Crude protein ‘%’	64.83 ^b^	63.83 ^b^	68.33 ^a^	0.642	0.003
Ether extract ‘%’	83.83	83.67	84.33	0.431	0.825
Crude fiber ‘%’	22.67	23.83	23.17	0.319	0.345

Means with different superscripts are significantly different; ^1^ SEM—standard error of means.

**Table 4 animals-11-00083-t004:** Effect of dry, wet and fermented wet feeds on carcass traits (%) of broiler chickens at 35 days of age.

Item	Dry Feed	Wet Feed	Fermented Wet Feed	SEM ^1^	*p*-Value
Dressing	70.92	70.45	70.69	0.447	0.923
Carcass yield	74.88	74.51	74.50	0.412	0.933
Liver	2.74	2.89	2.67	0.079	0.538
Gizzard	1.21	1.17	1.14	0.028	0.600
Heart	0.505	0.552	0.608	0.025	0.247
Spleen	0.117	0.147	0.173	0.015	0.307
Abdominal fat	1.41	1.18	1.20	0.083	0.479
Breast	23.84	22.17	23.90	0.537	0.352
Thigh	15.78	15.79	15.49	0.288	0.898

^1^ SEM—standard error of means.

**Table 5 animals-11-00083-t005:** Effect of dry, wet and fermented wet feeds on biochemical blood indices of broiler chickens at 35 days of age.

Item	Dry Feed	Wet Feed	Fermented Wet Feed	SEM ^1^	*p*-Value
Total protein ‘g·dL^−1^’	5.52	5.53	5.42	0.174	0.964
Albumin ‘g·dL^−1^’	1.55	1.52	1.62	0.043	0.653
Globulin ‘g·dL^−1^’	3.97	4.01	3.80	0.175	0.888
Albumin/globulin ratio	0.407	0.395	0.437	0.021	0.741
AST ‘U·L^−1^’	257.8	252.0	246.2	9.048	0.884
ALT ‘U·L^−1^’	5.50	6.00	4.32	0.560	0.479
Glucose ‘mg·dL^−1^’	162.3	161.2	154.7	7.904	0.922
Triglycerides ‘mg·dL^−1^’	60.17	55.83	55.33	4.407	0.898
Total cholesterol ‘mg·dL^−1^’	124.7	119.0	118.5	4.697	0.854
VLDL-cholesterol ‘mg·dL^−1^’	12.03	11.17	11.06	0.882	0.899

^1^ SEM—standard error of means.

## Data Availability

All data sets collected and analyzed during the current study are available from the corresponding author on fair request.

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
