# Peer review of "Effect of Feeding Wet Feed or Wet Feed Fermented by *Bacillus licheniformis* on Growth Performance, Histopathology and Growth and Lipid Metabolism Marker Genes in Broiler Chickens"

_animals, 2021, doi:10.3390/ani11010083_

Round 1

Reviewer 1 Report

Dear Authors,

This is quite an innovative study looking at effects of fermented wet diet on broiler growth. 

The following are my suggestions to the manuscript:

  1. The title is a bit confusing: I would recommend changing that to: (a). Effect of "Effect of Feeding Wet Feed Fermented by Bacillus ..." and (b). Bacillus Licheniformis in the title should be Bacillus licheniformis  
  2. Why was Bacillus licheniformis chosen over any other bacteria? there is no explanation for this in the introduction. The discussion section has some reference, but including this in the introduction would put this in context
  3. The use of superscripts (a,b) in the figures in not explained in any of the legends. Please replace these a,b symbols with statistical test numbers i.e. P-values, and please use the conventional asterisk method to denote all P-values observed. All P-value calculations must be described in the methods.  
  4. Are Figures 1,2 and 3 "representative" photomicrographs? If yes, this should be included in the legend. Also include the number of replicates (i.e. n) in every legend (method says n=8, but legend does not).
  5. Figure 4 and 5: Please replace "a" "b" with statistical P-values represented as asterisk. Please include number of replicates. 
  6. Line 243: "The investigation results .. are contradictory". Contradictory to what? I am assuming previous data, from reading the following lines.  Please detail/clarify.
  7. Line 257: "Scott agreed with our explanation". Is this a personal communication? If yes, this should be acknowledged. 
  8. Line 328: "These findings are consistent with the results obtained in this study". This sentence is technically wrong. Findings from one study cannot be consistent with the results from the same study. The results would have to be "concluded" based on the findings. Please revise. 
  9. The authors conclude that "fermented wet feed alleviates the harmful impact of heat stress" (Line 353). However, there is no data to support the alleviation of heat stress per se. Please include experimental evidence to support this claim. 

Author Response

Dear Prof. Editor-in-Chief, Animals

cc,

Prof.  Hannah Zhao, Assigned Editor

Regarding to the manuscript entitled "Effect of Feeding Fermented-Wet Feed by Bacillus Licheniformis on Growth Performance, Histopathology and Expression of Growth- and Lipid Metabolism-Related Genes in Broiler Chickens"          

Thank you in advance for your time and effort on reviewing our work.

A list of modifications according to the suggestions and comments of the reviewers is attached below. We are fully appreciated the valuable suggestions of the reviewers. Moreover, we are proud that our study has good discussion by the reviewers.

Sincerely Yours,

Ahmed Ali Mahmoud Saleh, PhD
Professor
Department of Poultry Production
Faculty of Agriculture,
Kafrelsheikh University, Egypt.

List of modifications according to the suggestions and comments of reviewers:

(Revisions related to reviewers’ comments are shown in red in the revised manuscript)

The authors appreciate the comments from the reviewers. The manuscript has been revised in accordance with their requests. We do our best to take all comments in consideration, incorporating them into the revised manuscript as indicated in our responses to the reviewers.

Comments and Suggestions for Authors

Author's Reply to the Review Report (Reviewer 1)

Dear Authors,

This is quite an innovative study looking at effects of fermented wet diet on broiler growth. 

The following are my suggestions to the manuscript:

  1. The title is a bit confusing: I would recommend changing that to: (a). Effect of "Effect of Feeding Wet Feed Fermented by Bacillus ..." and (b). Bacillus Licheniformis in the title should be Bacillus licheniformis  

Response: Thank you for your suggestion. We changed the title as your advice. As following (Effect of Feeding Wet Feed Fermented by Bacillus licheniformis   on Growth Performance, Histopathology and Growth and Lipid Metabolism Marker Genes in Broiler Chickens

  1. Why was Bacillus licheniformis chosen over any other bacteria? there is no explanation for this in the introduction. The discussion section has some reference, but including this in the introduction would put this in context

Response: Thank you for your suggestion.

We added more information in the introduction about Bacillus licheniformis  as following (Bacillus spp. seems to be an alternative to antimicrobial growth promoters for improving animals’ health and performance. Within this regard, Bacillus licheniformis have the ability to sporulate, thereby making them stable during thermal treatment of feed and resistant to enzymatic digestion along the gastrointestinal tract (GIT) [16]. Previous studies reported that Bacillus Licheniformis supplementation in broilers' drinking water or diets promoted the growth performance by enhanced the body weight and feed conversion ratio of broilers [6-12]) P2 L 71-76.

  1. The use of superscripts (a,b) in the figures in not explained in any of the legends. Please replace these a,b symbols with statistical test numbers i.e. P-values, and please use the conventional asterisk method to denote all P-values observed. All P-value calculations must be described in the methods.  

Response: Thank you for your suggestion.

We corrected it please, see the figures legends and the methods

  1. Are Figures 1,2 and 3 "representative" photomicrographs? If yes, this should be included in the legend. Also include the number of replicates (i.e. n) in every legend (method says n=8, but legend does not).

Response: Thank you for your suggestion.

We corrected it  Yes ,  we added it

  1. Figure 4 and 5: Please replace "a" "b" with statistical P-values represented as asterisk. Please include number of replicates. 

Response: Thank you for your suggestion.

We corrected it  

  1. Line 243: "The investigation results .. are contradictory". Contradictory to what? I am assuming previous data, from reading the following lines.  Please detail/clarify.

Response: Thank you for your suggestion.

We corrected it and added new sentences as following (One of the major objectives of the current study was to evaluate the impacts of feeding fermented feed on performance).P9 L248-249.

  1. Line 257: "Scott agreed with our explanation". Is this a personal communication? If yes, this should be acknowledged. 

Response: Thank you for your suggestion.

We corrected it as following (Scott and Silversides were agreed with our explanation, as the authors stated that the ability of broilers fed wet feed to utilize feed was decreased compared to those fed on dry feed [28].P10 L 263-264.

  1. Line 328: "These findings are consistent with the results obtained in this study". This sentence is technically wrong. Findings from one study cannot be consistent with the results from the same study. The results would have to be "concluded" based on the findings. Please revise. 

Response: Thank you for your suggestion.

We agreed with your advice and deleted the sentences.

  1. The authors conclude that "fermented wet feed alleviates the harmful impact of heat stress" (Line 353). However, there is no data to support the alleviation of heat stress per se. Please include experimental evidence to support this claim.

Response: Thank you for your suggestion.

We corrected it as following (Our results show that the fermentation of wet feed by B. Licheniformis addressed the disadvantages of wet feeding and improved protein digestibility. The use of fermented wet feed is more appropriate than the use of dry or wet feed as a potential approach to provide adequate nutrients to benefit from the genetic potential of broiler chickens' growth. P12 L356-359.

Reviewer 2 Report

The subject of this paper is of scientific interest and deals with the scope of the journal. The experimental design is appropriate, even though a higher number of replicates would be desirable. However, in my opinion, the ms is not adequate for publication in Animals as it is and needs deep review before being considered.

Reading the title of the ms it seems that the authors were only interested in evaluating the effect of fermented wet feed, however 2 different feed strategies (wet feed and fermented wet feed) were tested. Both should have the same weight when presenting results/discussion. Reading the Abstract/summary it seems that wet feed improved performance parameters, but this is not true (even numerically are worse that in controls). The following sentences are false: “Our findings confirm improved growth and digestibility for wet feed” (lines 28-29); “ fermenting the wet feed with Bacillus Licheniformis improved feed utilization and birds' performance compared to the dry feed group” (lines 48-49”. It is true that fermented liquid feed shows some positive effects on protein digestibility, gut histopathology and gene expression, but these were not translated into a better performance. When discussing the results, authors must clearly address these points and if they think that fermented wet feed offers advantages compared to dry feed at heat stress conditions must give strong arguments.

Specific comments

Line 50: to include wet feed

Line 65: efficiency

Line 89: ethical statement must be included

Line 119: please indicate how birds were selected

Line 170: more details are needed, indicating when pen/animal were the statistical units

Line 185: the title must refer also to digestibility

Line 193: “Means with different superscripts are significantly different” does not apply here.

Line 199: “VLDL: very low density lipoprotein” already stated previously

Line 256-257: The authors agree with Scott, not the contrary.

Line 264: the word “addressed” is not clear here and later on.

Line 284 (and so on): “insignificantly” better not significantly

Author Response

Dear Prof. Editor-in-Chief, Animals

cc,

Prof.  Hannah Zhao, Assigned Editor

Regarding to the manuscript entitled "Effect of Feeding Fermented-Wet Feed by Bacillus Licheniformis on Growth Performance, Histopathology and Expression of Growth- and Lipid Metabolism-Related Genes in Broiler Chickens"          

Thank you in advance for your time and effort on reviewing our work.

A list of modifications according to the suggestions and comments of the reviewers is attached below. We are fully appreciated the valuable suggestions of the reviewers. Moreover, we are proud that our study has good discussion by the reviewers.

Sincerely Yours,

Ahmed Ali Mahmoud Saleh, PhD
Professor
Department of Poultry Production
Faculty of Agriculture,
Kafrelsheikh University, Egypt.

List of modifications according to the suggestions and comments of reviewers:

(Revisions related to reviewers’ comments are shown in red in the revised manuscript)

The authors appreciate the comments from the reviewers. The manuscript has been revised in accordance with their requests. We do our best to take all comments in consideration, incorporating them into the revised manuscript as indicated in our responses to the reviewers.

Comments and Suggestions for Authors

Author's Reply to the Review Report (Reviewer 2)

Comments and Suggestions for Authors

The subject of this paper is of scientific interest and deals with the scope of the journal. The experimental design is appropriate, even though a higher number of replicates would be desirable. However, in my opinion, the ms is not adequate for publication in Animals as it is and needs deep review before being considered.

Reading the title of the ms it seems that the authors were only interested in evaluating the effect of fermented wet feed, however 2 different feed strategies (wet feed and fermented wet feed) were tested. Both should have the same weight when presenting results/discussion.

Response: Thank you for your suggestion.

We agreed with your observation, the wet feed products now using in different countries and has advantages and disadvantages so our objective focus for how can increase the utilization of the wet feed by different methods the fermentation method one of them.

 Reading the Abstract/summary it seems that wet feed improved performance parameters, but this is not true (even numerically are worse that in controls). The following sentences are false: “Our findings confirm improved growth and digestibility for wet feed” (lines 28-29); “ fermenting the wet feed with Bacillus Licheniformis improved feed utilization and birds' performance compared to the dry feed group” (lines 48-49”.

Response: Thank you for your suggestion.

We agreed with your observation and corrected as following (Our findings confirm improved growth and digestibility for fermented wet feed. Additionally, fermented wet feed caused a reduced hepatic gene expression of peroxisome proliferator-activated receptor γ (PPARγ) and increased that of fatty acid synthase (FAS).P1 L 29-30.

 It is true that fermented liquid feed shows some positive effects on protein digestibility, gut histopathology and gene expression, but these were not translated into a better performance. When discussing the results, authors must clearly address these points and if they think that fermented wet feed offers advantages compared to dry feed at heat stress conditions must give strong arguments.

Response: Thank you for your suggestion.

We corrected the conclusion as following (Our results show that the fermentation of wet feed by B. Licheniformis addressed the disadvantages of wet feeding and improved protein digestibility. The use of fermented wet feed is more appropriate than the use of dry or wet feed as a potential approach to provide adequate nutrients to benefit from the genetic potential of broiler chickens' growth. P12 L356-359.

Specific comments

Line 50: to include wet feed

Response: Thank you for your suggestion. We added it L50

Line 65: efficiency

Response: Thank you for your suggestion. We corrected it L65

Line 89: ethical statement must be included

Response: Thank you for your suggestion. We added it L96-98 as following (The study was approved by the Ethics Committee of Local Experimental Animals Care Committee and conducted in accordance with the guidelines of Kaferelsheikh University, Egypt (Number 4/2016 EC).

Line 119: please indicate how birds were selected

Response: Thank you for your suggestion.

We added it  as following (At the end of the experiment, two birds per replicate (eight birds each group) in the same body weight were slaughtered.) P4 L127-128.

Line 170: more details are needed, indicating when pen/animal were the statistical units

Response: Thank you for your suggestion.

We corrected it as following (Collected data were analyzed using a one-way analysis of variance, with General Linear Model's procedure, by the statistical analysis system SPSS 19 (2018). One-way ANOVA was applied to determine the effects of Wet feed, fermented wet feed, when birds were the statistical units for body weight and pens used for feed intake. To identify the presence of significance (P < 0.05) between means, Tukey's multiple comparison test was used.)

Line 185: the title must refer also to digestibility

Response: Thank you for your suggestion.

We added it as following (Table 3. Effect of dry, wet, and fermented-wet feeds on growth performance and digestibility of broiler chickens at 35 days of age.) L 195.

Line 193: “Means with different superscripts are significantly different” does not apply here.

Response: Thank you for your suggestion.

We deleted it as all the results are not significantly different

Line 199: “VLDL: very low density lipoprotein” already stated previously

Response: Thank you for your suggestion.

We deleted it

Line 256-257: The authors agree with Scott, not the contrary.

Response: Thank you for your suggestion.

We corrected it

Line 264: the word “addressed” is not clear here and later on.

Response: Thank you for your suggestion.

We deleted this sentence as the Reviewer 1advising.

Line 284 (and so on): “insignificantly” better not significantly

Response: Thank you for your suggestion.

We corrected it

-----------------------------------------------------------

Round 2

Reviewer 1 Report

Dear Authors,

Thank you for addressing my concerns. 

Author Response

Dear Authors,

Thank you for addressing my concerns. 

(x) English language and style are fine/minor spell check required

Response: Thank you for your suggestion.

We appreciate your comments however, we made the English editing for the manuscript before submitted.

Reviewer 2 Report

Even though the authors have addressed some of the points made in the first-round of review, the most important one is still unsolved.

If I have understood well the authors try to evaluate the effect of wet feeding to improve performance in broilers under heat stress ( Lines 61-62: “Wet feeding has been considered as a valuable tool to enhance birds' capability to endure high ambient temperatures and to increase our understanding of feed consumption limitations when broilers are fed dry diets”) and performed one experiment including two strategies (wet feed and fermented wet feed) to be compared versus the control dry-feed diet. Based on this, the results of both wet experimental diets must be compared against the control. In my opinion it is not correct to say: Our findings confirm improved growth and digestibility for fermented wet feed (lines 28-29) since NO differences were observed compared to the control. Moreover, the conclusion starts as “Our results show that the fermentation of wet feed by B. Licheniformis addressed the disadvantages of wet feeding…”, again the conclusion should start comparing the control dry-feed versus the experimental wet diets.

The title does not still make reference to wet food. Please include the “wet feed” treatment.

Author Response

Author's Reply to the Review Report (Reviewer 2)

Comments and Suggestions for Authors

Even though the authors have addressed some of the points made in the first-round of review, the most important one is still unsolved.

If I have understood well the authors try to evaluate the effect of wet feeding to improve performance in broilers under heat stress ( Lines 61-62: “Wet feeding has been considered as a valuable tool to enhance birds' capability to endure high ambient temperatures and to increase our understanding of feed consumption limitations when broilers are fed dry diets”) and performed one experiment including two strategies (wet feed and fermented wet feed) to be compared versus the control dry-feed diet. Based on this, the results of both wet experimental diets must be compared against the control. In my opinion it is not correct to say: Our findings confirm improved growth and digestibility for fermented wet feed (lines 28-29) since NO differences were observed compared to the control.

Response: Thank you for your suggestion.

We corrected it as following (Our findings confirm improved growth and digestibility for fermented wet feed compared with control and wet feed diets, however, wet feed has negative effect on performance compared with control group (dry-feed). L 29-31.

Moreover, the conclusion starts as “Our results show that the fermentation of wet feed by B. Licheniformis addressed the disadvantages of wet feeding…”, again the conclusion should start comparing the control dry-feed versus the experimental wet diets.

Response: Thank you for your suggestion.

We corrected it as following (Our results show that we feed has negative impact on performance compared with dry-feed however, the fermentation of wet feed by B. Licheniformis addressed the disadvantages of wet feeding and improved protein digestibility. The use of fermented wet feed is more appropriate than the use of dry or wet feed as a potential approach to provide adequate nutrients to benefit from the genetic potential of broiler chickens' growth. L 364-368.

The title does not still make reference to wet food. Please include the “wet feed” treatment.

Response: Thank you for your suggestion.

We corrected it as following (Effect of Feeding Wet feed or Wet Feed Fermented by Bacillus licheniformis on Growth Performance, Histopathology and Growth and Lipid Metabolism Marker Genes in Broiler Chickens)
